# Effectiveness of Biofeedback in Individuals with Awake Bruxism Compared to Other Types of Treatment: A Systematic Review

**DOI:** 10.3390/ijerph20021558

**Published:** 2023-01-14

**Authors:** Maryllian de Albuquerque Vieira, Ana Izabela Sobral de Oliveira-Souza, Gesa Hahn, Luisa Bähr, Susan Armijo-Olivo, Ana Paula de Lima Ferreira

**Affiliations:** 1Kinesiotherapy and Manual Therapeutic Resources Laboratory, Department of Physical Therapy, Federal University of Pernambuco, Recife 50740-560, Brazil; 2Faculty of Business and Social Sciences, University of Applied Sciences, Hochschule Osnabrück, 49076 Osnabrück, Germany; 3Faculty of Rehabilitation Medicine/Faculty of Medicine and Dentistry, University of Alberta, Edmonton, AB T6G 2R3, Canada

**Keywords:** awake bruxism, biofeedback, pain, anxiety, stress, oral health

## Abstract

Excessive masticatory muscle activity is generally present in awake bruxism, which is related to increased anxiety and stress. It has been hypothesized that biofeedback could potentially manage awake bruxism, however, its effectiveness has not been empirically analyzed in a systematic manner. Therefore, this systematic review was designed to determine the effectiveness of biofeedback compared to other therapies in adults with awake bruxism. Extensive searches in five databases looking for randomized controlled trials (RCTs) that included biofeedback to manage awake bruxism were targeted. The risk of bias (RoB) assessment was conducted using the Cochrane RoB-2 tool. Overall, four studies were included in this systematic review, all of which used the electromyographic activity of the masticatory muscles during the day and night as the main endpoint. Auditory and visual biofeedback could reduce the excessive level of masticatory muscle activity in a few days of intervention. The majority of the included studies had a high RoB and only one study had a low RoB. The standardization of the biofeedback protocols was also inconsistent, which makes it difficult to establish the ideal protocol for the use of biofeedback in awake bruxism. Thus, it is proposed that future studies seek to reduce methodological risks and obtain more robust samples.

## 1. Introduction

According to the International Consensus on the Evaluation of Bruxism [1], bruxism is defined as masticatory muscle activity, biting and/or grinding of the teeth at different times of the day, such as during sleep (sleep bruxism) and while awake (awake bruxism) [2,3]. Sleep bruxism is a rhythmic (phase) or non-rhythmic (tonic) muscular activity of the masseter and temporal muscles during sleep. On the other hand, awake bruxism is defined as repetitive or sustained contact of the teeth and/or sustained stiffness of the masticatory muscles with forced jaw movement, to the sides or forward, during the day [2]. Among the various factors involved in the etiology of Bruxism, higher levels of somatization, depression, anxiety and stress have been found as important factors in patients with awake bruxism, particularly in women [4]. Commonly, psychological aspects such as stress, anxiety, and depression are reported to be associated with bruxism [5,6,7].

The prevalence of awake bruxism is inaccurate and underestimated, as there is a lack of information as to the frequency of these events and how long they have been occurring [8]. Even so, through previous epidemiological studies, it is estimated that the prevalence rates are higher in adults and can vary from 10% to 13% for sleep bruxism and 22% to 31% for awake bruxism [9]. It is estimated that 85% to 90% of the population have reported episodes of bruxism during their lifetime [10]. During the COVID-19 pandemic, several studies investigated the correlation of psychosomatic symptoms with the presence of episodes of teeth clenching. In a population of 370 college students, 30.5% (n = 113) reported being “very stressed” during the pandemic, with low or very low sleep quality (44.3% (n = 164)), and 113 students reported clenching their teeth during the day, associated with headache [11].

Individuals with awake bruxism also commonly experience pain and decreased pain threshold in the masticatory and cervical muscles, headache, limitation of mandibular range of motion, sleep disorders, and general impairment of the oral health-related quality of life [12,13,14]. As numerous consequences are related to bruxism, different types of interventions have been adopted to minimize their effects, such as the use of botulinum toxin [15], occlusal splints [16], therapeutic exercises, acupuncture, electrotherapy, massage [10], and biofeedback therapy, among others [17,18]. Biofeedback therapy is a technology that has been used as a cognitive-behavioral approach to acting on the regulation of excessive muscle activity in subjects with sleep bruxism. By using visual and/or auditory biofeedback, individuals can readapt their muscles’ behavior, reducing excessive masticatory muscle activity [2,18,19]. Biofeedback therapy has been shown to benefit patients with bruxism to relieve symptoms, preventing oral complications (i.e., destruction of teeth and restorations) [20], induce changes in quality of life, decrease levels of anxiety and stress [21], reduce muscle activity, and reduce pain for a prolonged period of time [17,22,23,24]

Despite the positive effects of biofeedback treatment in patients with orofacial symptoms, this intervention modality has not been equally explored in both types of bruxism (i.e., sleep and awake bruxism). Although three previous reviews have been published looking at the effect of biofeedback on bruxism, they presented some methodological problems. Two reviews [17,23] included only sleep bruxism as the main focus, which is an important flaw of these reviews. The only review that has searched for therapies for patients with awake bruxism has not investigated the effects of biofeedback therapy, but rather therapies in a general way such as electrotherapeutic, cognitive-behavioral therapy, therapeutic exercises, acupuncture, postural awareness, muscular relaxation, and massage [9]. In addition, two reviews only included the English language [9,23], and the search was conducted in only two databases (National Library of Medicine’s Medline and Scopus). These are important weaknesses of these reviews that need to be addressed as systematic reviews need to be comprehensive, including several languages, be up to date, and follow a strict methodology to avoid selection biases [23]. In addition, the main treatment focus in these reviews was physiotherapy interventions [9], and biofeedback treatment was not explored exhaustively. Thus, it was not possible to isolate the effects of biofeedback in the treatment of people with awake bruxism. Therefore, the effects of biofeedback therapy should be looked at for awake bruxism, specifically to guide clinicians and researchers in the use of this technique for this group of patients, which is growing rapidly.

Based on the above information, it was clear for our team that a new systematic review was necessary to address these weaknesses and to fill in this gap in the literature to provide clearer information about the effectiveness of biofeedback for awake bruxism [20,25]. Therefore, the aim of the present systematic review was to compile and synthesize the information about the effectiveness of biofeedback in the management of awake bruxism in the adult population, compared to other types of treatment.

## 2. Materials and Methods

### 2.1. Protocol and Registration

The protocol for this systematic review was registered in the international prospective register of systematic reviews, PROSPERO (https://www.crd.york.ac.uk/PROSPERO/, accessed on 30 November 2022), register number: CRD42021227084.

The question of interest for this review is: “what is the effectiveness of biofeedback on masticatory muscle activity in adults with awake bruxism when compared to other types of treatment?” We used the PICOS framework to organize our review:

P = “adults with awake bruxism”, I = “biofeedback”, C = ”any other type of intervention”, O = “masticatory muscle activity”, and S = “RCT studies”.

### 2.2. Search Strategy

This systematic review was part of a larger project looking at physical therapy strategies to manage bruxism, and biofeedback was included as one of these strategies. The searches were conducted by a team (MAV, LB, GH, AISOS) on 1 October 2021 and updated on 12 July 2022, in the following five databases: EBSCO/CINAHL, Embase, PubMed/Medline, Cochrane library (Wiley Interface), Web of Science (Indexes = SCI-EXPANDED, SSCI, A&HCI, ESCI). The searches included all possible terms and referents to bruxism based on the new nomenclature suggested by Lobbezoo, et al. in 2018 [1]. The final version included an extensive list of keywords about bruxism, biofeedback, and other physiotherapy modalities. The search was limited to controlled and randomized clinical trials including humans, and no time or language restrictions were applied. Manual searches were conducted by reviewing the reference list of articles included in this review. In order to identify new studies, a Scopus screen was performed. The whole search strategy is described in Appendix A.

### 2.3. Eligibility Criteria

#### 2.3.1. Inclusion Criteria

This systematic review included Controlled Clinical Trial (CCT) and Randomized Controlled Trials (RCT) studies involving adults (18–60 years) of both sexes with awake bruxism, diagnosed through physical and/or clinical evaluation performed by a dentist or a specialized health practitioner. If the study included sleep bruxism and the effects of awake bruxism could be isolated, the study was included. In addition, studies should investigate the effectiveness of biofeedback using auditory and/or visual signals compared to any type of conservative and non-conservative therapy, such as dental treatments (e.g., oral rehabilitation), physiotherapy (any modality), manual therapy, exercise therapy, placebo, occlusal splint, psychological intervention, pharmacotherapy, cognitive behavioral therapy, control group, among others.

Masticatory muscle activity and bruxism events (phasic and/or tonic events) were considered the primary outcomes of this review. Both can be measured by EMG (in microvolts), preferably during the daytime. EMG is a technology for evaluating or monitoring neuromuscular behavior. The signal is captured from the electrical potentials that the muscles emit during an activity [26].

The secondary outcomes were considered to be pain intensity (e.g., Visual Analog Scale, Numerical Rating Scale, among others); quality of life (e.g., Short Form 36 health survey, Oral Health Impact Profile 14, among others); mandibular function (e.g., Jaw Functional Limitation Scale, Mandibular Functional Impairment Questionnaire, among others); mandibular range of motion (e.g., measured with a ruler, caliper, others); and psychologic aspects (e.g., Hospital Anxiety Depression Scale, Beck Depression Inventory, among others).

#### 2.3.2. Exclusion Criteria

Studies that included children (<18 years) or elderly (>60) patients, or patients with other diseases such as Neurological, rheumatic, vascular, metabolic disease; cancer; neuropathic pain conditions; gastroesophageal reflux exclusively, diagnosis of sleep bruxism exclusively; use of biofeedback in people with exclusively other orofacial diagnoses (i.e., TMD, orofacial pain, headache) but in absence of awake bruxism, and previous surgery in the orofacial region were excluded. If the biofeedback treatment was combined with other therapies, but its effect could not be isolated, the study was excluded. Clinical trial protocols, cross-sectional studies, case-controlled, case studies, prospective studies, reviews (narrative of systematic), qualitative studies, commentaries, and letters to the editor were also excluded. However, potential references were screened for inclusion.

### 2.4. Study Selection

The research results were compiled into an ENDNOTE database and then imported into Covidence (www.covidence.org, accessed on 30 November 2022), which is the platform used for screening studies. The PRISMA flowchart [27] was used to organize all of the studies that were duplicated, and these were selected and removed. Two independent reviewers screened the titles and abstracts in the first step, and then full texts, taking into consideration the previously established inclusion and exclusion criteria. If conflicts occurred during the selection process, a consensus meeting was held with a third reviewer. The third reviewer (senior author of this review-SAO or APL) was an arbitrator to reach a consensus when needed.

### 2.5. Data Extraction

The data extraction form was created in a Microsoft Excel spreadsheet (Microsoft Corporation 2007). Independent reviewers extracted each study’s information (MAV, GH, LB). A second reviewer (AISOS) checked the data for consistency and completeness. The data extracted included information about objectives, study design, recruitment, allocation of participants, diagnostic criteria for awake bruxism, and details of the comparison groups and outcomes, as well as descriptions and details of biofeedback therapy (type of feedback, duration, frequency of intervention, and follow-up of participants), the number of participants in each phase (losses and withdrawals), and the main and secondary outcomes with tools used for evaluation. When applicable, the quantitative results were reported as mean and SD, median, and interval interquartile, confidence interval (CI), based on the pre-and post-treatment results between groups. Conclusions related to the outcomes of each study were also extracted.

### 2.6. Quality Assessment

The revised Cochrane risk-of-bias tool for randomized trials (RoB 2) was used to assess the risk of bias (RoB) in the randomized trials [28]. RoB 2 is structured into a fixed set of domains of bias, where each domain is a series of questions (‘signaling questions’) that aim to obtain information on the risks of bias in each study. Judgment can be ‘Low’, ‘Some concerns (Unclear)’ or ‘High’ risk of bias, described as: High risk of bias: if the study has a high RoB in at least one domain; Unclear risk of bias: if the study is unclear in at least one domain, and Low risk of bias: if the study has a low risk of bias in all domains. In cases of disagreement, the reviewers resolved by consensus (MAV, GH, LB). If no agreement was achieved, an independent third reviewer (AISOS) was invited to resolve and make the final decision.

### 2.7. Quality of Evidence

The level of certainty of the evidence was assessed through the GRADE (Grading of Recommendations, Assessment, Development, and Evaluation) system [29]. This grading system classifies the results in a level of evidence as high, moderate, low, and very low evidence considering five domains: study limitation, inconsistency, imprecision, indirectness, and publication bias [30].

### 2.8. Synthesis of Results

The synthesis of the data was in a narrative form. The information from the studies included in this review was organized in tables, with a summary of the results. The presentation of the data was based on the outcomes and comparisons between the groups of interest. The details about the population characteristics (gender, age, diagnosis), the comparison groups (biofeedback vs. other interventions), and the results of the outcomes analyzed by the studies (muscle activity, pain) can be found in the descriptive tables.

Statistical analysis was performed using the software Review Manager (RevMan) version 5.3. The biofeedback group was compared with the control group, which did not receive treatment; the outcomes were tonic and phasic muscle activity during the day and night. The standardized mean difference (SMD) was used to analyze the continuous variables with a 95% confidence interval (CI). The heterogeneity was calculated by I^2^, classified as 0% to 40%: may not be important; 40% to 60%: may represent moderate heterogeneity; 60% to 90%: may represent substantial heterogeneity; 90% to 100%: considerable heterogeneity as stated in the Cochrane handbook [31]. The results of the risk of bias (RoB2) and the level of evidence quality of the studies (GRADE) are represented descriptively and condensed in tables for better analysis.

## 3. Results

### 3.1. Study Selection

As presented in the PRISMA figure, initially 4059 studies were identified in the databases; of these, 2395 studies were duplicates. Based on the titles and abstracts, 86 studies were included in the full-text review. Eighty-two studies were excluded for different reasons: twenty-one (25%) did not include individuals with bruxism; nineteen (23%) did not use biofeedback as treatment; eleven (13%) were not clinical trials; eleven (13%) included individuals with only sleep bruxism; eight (9%) were abstracts; and twelve (14%) studies were clinical trial protocols (Figure 1). The list of excluded studies and reasons is available upon request from the authors. Thus, in total, four studies were included in this systematic review.

### 3.2. Studies Characteristics

The characteristics of the four included studies are shown in Table 1. A total of 69 adults of both sexes, diagnosed with awake bruxism [32,33] and with sleep bruxism were analyzed by this review [24,34]. Three studies (75%) compared biofeedback with the control groups (no intervention) [16,17,26] and one [34] compared biofeedback versus a Myomonitor-training (TENS) group (25%). The outcome evaluated by these studies was masticatory muscle activity with EMG. All of the biofeedback therapies used in the included studies were portable EMGs: three (75%) with auditory feedback [24,32,33] and one (25%) with visual feedback [34]. The parameters of the signals captured from the anterior temporalis and masseter muscles were in microvolts. The description of the intervention groups with biofeedback is presented in Table 1.

The main outcome evaluated by the studies was the EMG activity of the masticatory muscles during the day and night, also presenting information on the type of muscle contraction: phasic or tonic activity. The phasic contraction was considered when the EMG activities exceeded 10% MVC (maximal voluntary contraction) for 1 s and tonic when there were sustained elevations of muscular activity for more than 2 s. Both outcomes were measured in microvolts for 20 min to 5 h of intervention. The description of the post-intervention results is presented below, comparing biofeedback versus comparison groups.

### 3.3. Risk of Bias

The RoB of the included studies was variable; two of them presented a high RoB [32,33], one presented unclear RoB [34], and one study presented a low RoB [24]. Based on the domains, in Domain 1 (randomizing process), two (50%) studies presented a high RoB [32,33], and two (50%) had some concerns [24,34]. Regarding the deviations from the intended intervention’s domain, all of the studies (100%) had some concerns in this domain (Figure 2) [32,33].

### 3.4. Effectiveness of the Biofeedback Treatment

#### 3.4.1. Electromyography Activity of the Masticatory Muscles

##### Auditory Biofeedback vs. Control Group (Non-Intervention)

Three (75%) studies [24,32,33] compared the effectiveness of biofeedback to the control group with no intervention. The results showed that there was a significant difference between groups on daytime tonic events (SMD = −1.31 [95%CI −2.10, −0.51], *p* = 0.001). Regarding daytime phasic events (under the 10%MVC threshold for 1s), a significant difference was also found between auditory biofeedback vs. the control group (SMD = −1.92 [95%CI −2.80, −1.03], *p* < 0.0001) (Figure 3). Both outcomes presented a very low level of evidence based on the GRADE, with low heterogeneity in the tonic and phasic daytime events (I^2^ = 0%).

When comparing auditory feedback versus a control group, a significant and positive effect favoring the auditory biofeedback group on night time tonic events was found (SMD = −1.39 [95%CI: −2.72, −0.06] *p* = 0.04). However, only one study [32] investigated that outcome (night time tonic event); thus, the meta-analyses could not be performed. Similar results were obtained for night time phasic events, also favoring the biofeedback group (SMD = −1.66 [95%CI: −2.56, −0.77]; *p* = 0.0003), with a low heterogeneity (I^2^ = 0%) (Figure 3). This result had a very low level of evidence based on the GRADE for both outcomes (Table 2).

##### Visual Biofeedback vs. Myomonitor (TENS)

Only one (25%) study [34] compared visual biofeedback to transcutaneous electrical neuromuscular stimulation (TENS). The results were favorable for the biofeedback group in reducing the EMG activity of the left masseter (SMD = −1.24 [95%CI: −2.21, −0.26]; *p* = 0.01), right temporalis (SMD = −1.06 [95%CI: −2.01, −0.11]; *p* = 0.03), and left temporalis (SMD = −1.16 [95%CI: −2.12, −0.19]; *p* = 0.02) muscles, while there was no difference between groups in relation to the right masseter muscle (SMD = −0.55 [95%CI: −1.45, 0.34], *p* = 0.23) (Figure 4). This result had a low level of evidence based on the GRADE (Table 3).

#### 3.4.2. Myofascial Pain Intensity

Although the study by Watanabe et al. 2011 [33] proposed to investigate the intensity of pain with the numerical rating scale (NRS) in individuals with mild to moderate pain, the results related to pain intensity were not reported after the end of the intervention. Thus, no data in the previous literature were provided related to the pain intensity outcome.

#### 3.4.3. Other Outcomes

Although other outcomes were considered for this review, none of the included studies investigated the quality of life, mandibular function, range of motion, or psychological aspects.

### 3.5. Quality of Evidence

The qualitative evidence of all of the studies was performed using the GRADE and is presented in Table 2 and Table 3. The quality of the evidence of the studies’ outcomes ranges between very low (i.e., daytime, and nighttime muscle activity) and ‘low’ (i.e., right, and left masseters and temporalis). The tonic and phasic events (daytime and nighttime) present a serious RoB due to the high risk of bias present in the RoB-2 domains. The inconsistency of outcomes was serious for the nighttime tonic activity outcome and very serious for the other outcomes [31]. While the indirectness was not serious, the imprecision was serious in all of the studies because the number of participants was small (n < 300).

## 4. Discussion

### 4.1. General Results

The main finding of this systematic review is that there was a reduction in the tonic and phasic events of the masticatory muscles during the day and the night in subjects with awake bruxism treated with auditory biofeedback (BF-A), when applied for two consecutive days for one week. In addition, visual biofeedback (BF-V) reduced the EMG activity in the left masseter and both temporalis muscles in individuals treated once a week for three weeks with 20 min sessions, when compared to TENS treatment.

### 4.2. Effectiveness of Biofeedback

The presence of phasic activity of the masticatory muscles affects at least 16.8% of individuals with awake bruxism, and high tonic activity can also be found in this population (32.3%) [35]. The repetitive and/or sustained activity of the masticatory muscles can result in tooth wear and softening pain [36], as well as increased muscle tension [37]. Therefore, the reduction in these events by biofeedback would potentially reduce and prevent injuries caused by repetitive tooth clenching, such as in awake and sleep bruxism.

The effectiveness of biofeedback is supposedly related to a relationship between tooth clenching and the activation of the prefrontal areas of the brain [38,39], which are the areas responsible for regulating the motor activity of the trigeminal nerve, particularly in situations when an emotional response is required. Therefore, when faced with an event that triggers stress (emotional) reactions, chewing activities (i.e., clenching) become a pleasurable activity in response to the lower level of serotonin and high level of stress [40,41]. Thus, when biofeedback therapy has a positive effect on reducing masticatory activity, it is supposed that, in consequence, biofeedback promotes the relaxation of tense masticatory muscles through awareness of the habit of clenching teeth and the reduction in stress (emotional) level, which can directly affect the symptoms of bruxism, which are closely related to stress levels [42].

Although biofeedback has a positive effect on reducing symptoms in patients with bruxism, there is no consensus concerning which is the best biofeedback protocol to use, particularly in regard to how much time is needed to generate a change in the activity of the affected muscles and to decrease bruxism symptoms [43]. The literature showed that only two days of therapy using auditory biofeedback were effective enough to decrease the tonic and phasic muscle activity of these patients when compared to no treatment [24,32,33]. However, when the visual biofeedback was applied just once a week, no reduction in the EMG activity levels of the masticatory muscles was found in this systematic review [34].

In addition, when biofeedback is used for a longer/prolonged period of time, which is considered more than four weeks of treatment, it can generate a “habituation” of the received signal, which means that muscles become used to the stimulus and no more activation could be seen [44,45]. That is, there is habituation with the received signal (auditory or visual) and the decrease in muscle activity remains unchanged because the individual has adapted to the feedback used in the treatment. At this point, if more training is needed to promote further muscle relaxation, it will probably be necessary to change the signal type or the therapy provided. Although the results from this SR could not strongly conclude the effectiveness of biofeedback for awake bruxism due to the small number of studies included, it is important to highlight that the number or frequency of biofeedback sessions could be a factor that potentially affected the results of this therapy.

### 4.3. Previous Reviews

Previous reviews have provided different results regarding the effectiveness of biofeedback for sleep bruxism and none of them included individuals with awake bruxism. Therefore, to the best of our knowledge, this is the first review that summarizes this evidence. A previous review looking at sleep bruxism [17] concluded that contingent electrical stimulation (CES) biofeedback can reduce muscle activity in sleep bruxism in the short term (immediately after the end of the treatment). While Wang et al. [18] showed that CES has no effect on sleep bruxism, the other two reviews reported that biofeedback can be an option for the management of sleep bruxism [9,23]. However, all of those reviews did not fully support the effectiveness of biofeedback due to the low number of studies included and the low quality of the evidence. Therefore, in this review, it is suggested that better clinical trials with larger sample sizes and long-term follow-up are conducted.

### 4.4. Quality of Evidence and Risk of Bias

In general, the quality of the analyzed studies was poor. Only one study [24] had a low risk of bias overall. While looking at the domains individually from the RoB tool, it is noticed that several methodological flaws were evident. For example, in the majority of the studies, there was uncertainty about the blinding of subjects or assessors. As the outcome of the analyzed studies was EMG activity, the lack of blinding of the assessors may be associated with detection biases and could thus have a negative impact on the study results.

In addition, most of the studies did not include appropriate randomization and allocation concealment procedures, even among those that presented good randomization, the intent-to-treat (ITT) analysis was not followed. All of these methodological flaws can result in bias in the results. It has been presented by the literature that when no ITT analyses were used and, consequently, the randomization benefits are not maintained in the study, the results from a clinical trial could be overestimated [46,47]. This implies that the results presented by this review may be overestimated because of the biases presented by the included studies.

### 4.5. Considerations for Research and Clinical Practice

Based on the results from this systematic review, physical therapists and other health professionals who work in the orofacial area could better understand the role of biofeedback in the management of awake bruxism in the short term. As it is anticipated that cases of clenching during the day will increase due to the current stressful lifestyle of the population, interventions that act directly on this complaint are needed. In addition, we must take into consideration other factors that are related to awake bruxism, such as stress and/or anxiety levels, chronicity, and the interest of the individual in participating in the treatment. Therefore, we recommend that professionals evaluate these outcomes and consider the patient’s needs.

The included studies only considered masticatory muscle activity as an outcome, measured with EMG. Although this is one of the most important clinical signs in the detection of awake bruxism, individuals with awake bruxism can also report other clinical manifestations, such as masticatory and cervical muscle pain, headaches, increased stress levels, anxiety, and poor oral health quality. These clinical outcomes were not explored by the analyzed studies; therefore, future research should investigate them specifically.

Considering that methodological flaws may interfere with the results of a study, as well as when summarizing the results in a review, this current systematic review suggests that future studies should try to minimize biases when designing and conducting studies in this area such as performing appropriate randomization and allocation concealment and attempting to blind outcome assessors, when possible, specifically when outcomes are assessor-performed/-dependent. In addition, based on the literature analyzed, no clear protocol for treatment has emerged; thus, the question remains about the appropriate protocol for the treatment of awake bruxism with biofeedback.

Furthermore, no other review has attempted to explain the effectiveness of biofeedback in individuals with awake bruxism. Thus, this review was extremely necessary to clarify the studies that have been conducted on this topic to date.

### 4.6. Strength and Limitations

This systematic review sought to include all terms related to awake bruxism to ensure that all possible studies related to the topic were included in the search. The reviewers’ collaboration was also essential to maintain methodological rigor. The limited number of studies included in this review was due to the fact that there are not enough studies in the literature that focused on awake bruxism. The limited number of studies included and their heterogeneity made it difficult to compile their results and provide a better meta-analysis. Moreover, three of the four included studies are from the same department, which could represent a high risk of bias for the results of this systematic review as systematic errors could be magnified.

## 5. Conclusions

Biofeedback seems to have the potential to reduce the level of masticatory muscle activity in individuals with awake bruxism at short-term follow-up. However, the small number of studies published on this topic, and the low level of evidence from the studies, added to the high risk of bias presented by them, do not fully support the effectiveness of biofeedback for awake bruxism. It is therefore recommended that clinical trials have larger sample sizes and be conducted with high methodological standards to avoid important bias associated with inappropriate randomization, blinding, and deviations from intended interventions, as found in this review. Furthermore, it is recommended that, in addition to using muscle activity as the main outcome, other clinically relevant outcomes such as pain, quality of life, and disability, among others, are used not only in short-term but also at long-term follow-up for this group of patients.

## Figures and Tables

**Figure 1 ijerph-20-01558-f001:**
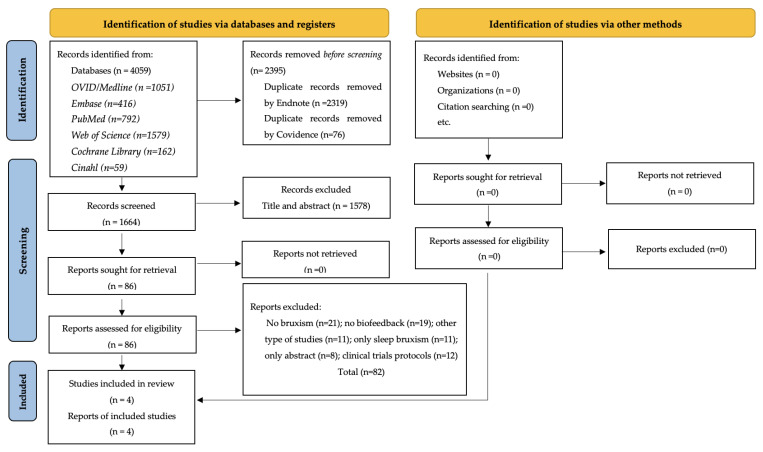
PRISMA Flowdiagram.

**Figure 2 ijerph-20-01558-f002:**
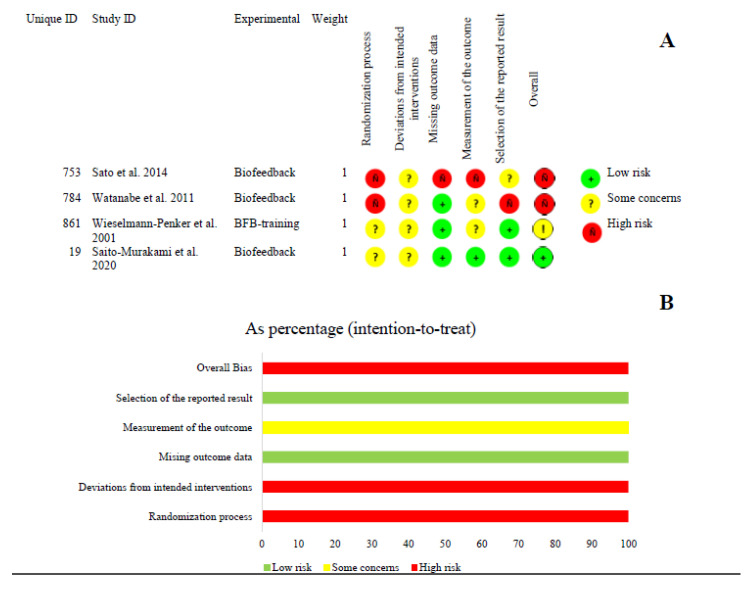
Risk of bias–Rob2. (**A**) The risk of bias by domain is presented in five domains (columns) and specified by studies (lines). (**B**) Risk of bias is presented for all studies combined as percentage based on the frequency of low, unclear, and high risk of bias. (The intention-to-treat approach was used as the basis for this analysis) [24,32,33,34].

**Figure 3 ijerph-20-01558-f003:**
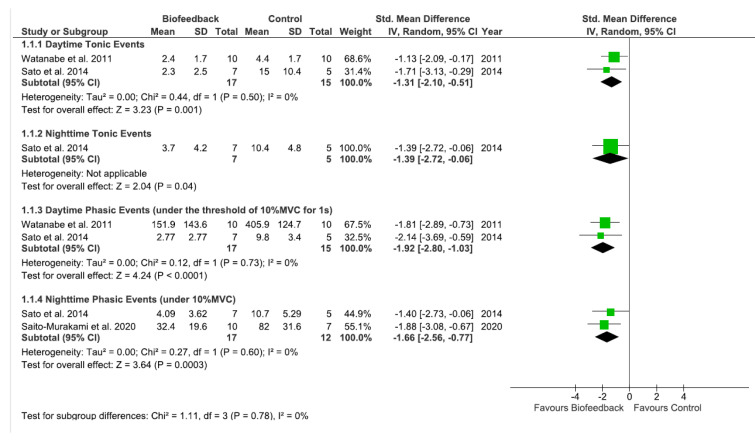
Day and night times phasic and tonic events of bruxism activity: Biofeedback training vs. control groups evaluated through EMG activity [24,32,33].

**Figure 4 ijerph-20-01558-f004:**
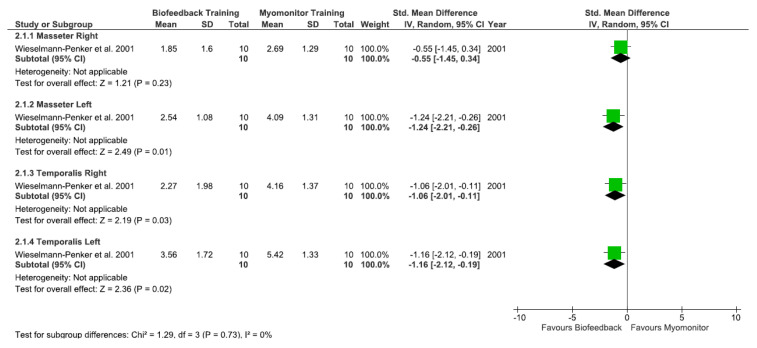
EMG activity per muscle: Visual feedback vs. Myomonitor (TENS) evaluated through EMG activity [34].

**Table 1 ijerph-20-01558-t001:** Summary of the studies’ characteristics.

Study	Population Characteristic	Biofeedback Parameters	Treatments Information	Outcomes-Tool	Results Summary after Treatment (Mean ± SD)
Sato et al., 2014 [32] **Country:** Japan **Aim:** To determine the effect of EMG BF for AB tonic EMG events on SB tonic EMG events, to expand new concepts of SB regulation.	**Age:** 26.8 ± 2.5 years **Sex:** Male **Diagnosis:** awake bruxism **Diagnosis tool:** Self-reported, Clinical assessment **Diagnosis duration:** No reported **Total sample size:** 12 **Number of groups:** 2	**Type of BF:** Audio EMG portable Application area: Temporal muscle (habitual side of chewing)	BF Group **Sample (n):** 7 **Study design timeline:** 3 weeks. Week 1 and 3 of evaluation pre and post-intervention. Week 2–treatment. **Number of sessions:** 2 **Duration:** 2 consecutive days of daytime training and 5 h of recording at night in the second week. Control Group (no intervention) **Sample (n):** 5 **Duration:** 5 h of recording (day/night) in 2 days in the second week.	1. **Bruxism activity–EMG analysis** 1.1.Tonic events (daytime) 1.2. Tonic events (nighttime) 1.3. Phasic events (daytime) 1.4. Phasic events (nighttime)	Outcome 1.1. The number of tonic events decreased after 2 days of treatment (2.3 ± 2.5 RMS) in the BF group but was not statistically significant. In the control group, the values were unchanged. Outcome 1.2. No information about results between groups. Within groups, there was a decrease in week 3 only in the BF group (3.7 ± 4.2 RMS). Outcomes 1.3. and 1.4. No comparison between groups was provided. Within groups, no significant difference was found in both groups.
Watanabe et al., 2011 [33] **Country:** Japan **Aim:** To ascertain the effect of EMG BF on the regulation of daytime clenching behavior.	**Age:** 30.9 ± 6.8 years **Sex:** Mixed **Diagnosis:** awake bruxism **Diagnosis Tool:** Self-reported, Clinical assessment **Diagnosis duration:** No reported **Total sample size:** 20 **Number of groups:** 2	Type of BF: Audio EMG portable Application area: Temporalis muscles anterior part	BF Group **Sample (n):** 10 **Study design timeline:** 1 week. Day 1 and day 4–pre and post treatment. Days 2 and 3–treatment. **Number of sessions:** 2 **Duration:** 2 consecutive days of daytime training. Control Group (no intervention) **Sample (n):** 10 **Study design timeline:** same of BF group, but without feedback signal. **Duration:** 5 h of recording in daytime in 2 consecutive days.	**1. Bruxism activity-EMG** 1.1. Phasic events (daytime) 1.2. Tonic events (daytime) **2. Pain-NRS**	Outcome 1.1 On day 4, after 2 days of training, fewer EMG activities (phasic events) were found in the BF group (151.9 ± 143.6 RMS) than in the CO group (405.9 ± 124 RMS. *p* < 0.05). No information about between groups comparison. Outcome 1.2. A significant decrease in the number of clenching events (tonic events) was found in the BF group between Day 1 (4.6 ± 2.5 RMS) and Day 4 (2.4 ± 1.7 RMS; *p* < 0.05). There was no significant change in clenching events in CG. No information about the comparison between groups was provided. Outcome 2. No information post intervention.
Wieselmann-Penker et al., 2001 [34] **Country:** Austria **Aim:** To compare EMG BF training and myomonitor training (TENS) in different kinds or degrees of EMG activity in the masticatory muscles at rest position.	**Age:** 22 and 58 years **Sex:** Mixed **Diagnosis:** Bruxism **Diagnosis Tool:** Clinical assessment **Diagnosis duration:** 6 months **Total sample size:** 20 **Number of groups:** 2	Type of BF: Visual EMG portable Application area: Masseter muscles and Temporalis muscles (bilaterally)	BF Group **Sample (n):** 10 **Study design timeline:** weekly for 3 times **Duration:** 20 min of relaxing masticatory muscles (10 min each for masseter and temporalis) by observing visual feedback. **Number of sessions:** 3 MM Group (TENS) **Sample (n):** 10 **Study design timeline:** weekly for 3 times. **Duration:** 20 min of rest. **Number of sessions:** 3	1. **Bruxism activity–EMG** 2. **SCL-EMG**	Outcome 1. EMG levels of session I, II and III indicated a significant between-group effect not only at pre- and post-treatment (*p* = 0.002) analysis of both masseter muscles but also at post-treatment measuring of the right temporalis. Outcome 2. Analysis of mean EMG levels and SCL during baseline screening did not yield any statistically significant differences for the MM and BFB groups.
Saito-Murakami et al., 2020 [24] **Country:** Japan **Aim:** To determine the effect of daytime EMG BF on RMMA during sleep in individuals with bruxism.	**Age:** 23.9 ± 3.3 years **Sex:** Male **Diagnosis:** AB and SB **Diagnosis Tool:** Self-reported, Clinical assessment **Diagnosis duration:** No reported **Total sample size:** 17 **Number of groups:** 2	Type of BF: Audio EMG portable Application area: Temporalis muscles anterior part	BF Group **Sample (n):** 10 **Study design timeline:** 3 weeks. Being week 1 and 3 of evaluation pre and post intervention. Week 2–treatment. **Duration of treatment:** 2 consecutive days of daytime training and recording at night in second week per 5 h. **Number of sessions: 2** Control Group (no intervention) **Sample (n):** 7 **Study design timeline:** Same of BF group. **Duration:** 2 consecutive days of recording without signal, during 5 h (day and night).	1. **Bruxism activity–EMG** 1.1. Tonic events (daytime) 1.2. Tonic events (nighttime) 1.3. Phasic events (daytime) 1.4. Phasic events (nighttime) 2. **Body movements during sleep–PSG and video recording**	Outcome 1. In week 3, phasic events were found to be significantly different between the BF (32.4 ± 19.6 RMS; mean ± SD) and the CO group (82.0 ± 31.6 RMS; mean ± SD; *p* = 0.006). Outcome 2. The maximum number of body movements overnight was 88, the minimum 54 and the average was 74.2. These results indicate that 7.8% of the total number of events was not actual EMG events caused by bruxism; 92.2% of the events were estimated as actual RMMA.

SD: standard deviation; RMS: root mean square; AB: Awake bruxism; SB: Sleep bruxism; BF: Biofeedback; MM: Myomonitor; CG: Control Group; EMG: electromyography; NRS: Numeric Rating Scale; SCL: skin conduct level; PSG: polysomnography; RMMA: rhythmic masticatory muscle activity.

**Table 2 ijerph-20-01558-t002:** GRADE “Biofeedback compared to control intervention for Awake Bruxism”.

Certainty Assessment							N° of Patients		Effect	Certainty
N° of Studies	Study Design	Risk of Bias	Inconsistency	Indirectness	Imprecision	Other Considerations	Biofeedback	Control Intervention	Absolute (95% CI)	
*Daytime Tonic Events (follow-up: mean 1–2 weeks; assessed with: EMG; Scale from: 0 to 300 mv)*										
2	randomised trials	very serious ^a^	not serious ^b^	not serious	serious ^c^	none	17	15	SMD **1.31 SD** **lower** (2.1 lower to 0.51 lower)	⨁◯◯◯ Very low
*Nighttime tonic activity (follow-up: mean 1–2 weeks; assessed with: EMG; Scale from: 0 to 200 mv)*										
1	randomised trials	very serious ^a,d^	serious ^a,d^	not serious	serious ^a,e^	none	7	5	SMD **1.39 SD** **fewer** (2.72 fewer to 0.06 fewer)	⨁◯◯◯ Very low
*Daytime Phasic Events (under the threshold of 10%MVC for 1s) (follow-up: mean 1–2 weeks; assessed with: EMG; Scale from: 0 to 300 mv)*										
2	randomised trials	very serious ^a^	not serious ^b^	serious ^c^	serious ^c^	none	17	15	SMD **1.92 SD** **lower** (2.8 lower to 1.03 lower)	⨁◯◯◯ Very low
*Nighttime Phasic Events (under the threshold of 10%MVC for 1s) (follow-up: mean 1–2 weeks; assessed with: EMG; Scale from: 0 to 300 mv)*										
2	randomised trials	very serious	not serious ^b^	not serious	serious ^c^	none	17	12	SMD **1.66 SD** **lower** (2.56 lower to 0.77 lower)	⨁◯◯◯ Very low

**CI:** confidence interval; **SMD:** standardized mean difference; ^a^ the studies present “High risk” in rob2; ^b^ I^2^ is less than 75%;^c^ the number of the total participants was under 300 (105); ^d^ the number of the total participants was under 300 (105). ^e^ the number of the total participants was under 300 (105).⨁◯◯◯ = very low evidence level; ⨁⨁◯◯ = low evidence level; ⨁⨁⨁◯ = moderate evidence level; ⨁⨁⨁⨁ = high evidence level.

**Table 3 ijerph-20-01558-t003:** GRADE “Biofeedback compared to Myomonitor Training for Awake Bruxism”.

Certainty Assessment							N° of Patients		Effect	Certainty
N° of Studies	Study Design	Risk of Bias	Inconsistency	Indirectness	Imprecision	Other Considerations	Biofeedback	Control Intervention	Absolute (95% CI)	
*Masseter Right (follow-up: mean 3 weeks; assessed with: EMG; Scale from: 0 to 10)*										
1	randomised trials	Serious ^a^	not serious	not serious	serious ^b,c^	none	10	10	SMD **0.55 SD** **lower** (1.45 lower to 0.34 higher)	⨁⨁◯◯ Low
*Masseter Left (follow-up: mean 3 weeks; assessed with: EMG; Scale from: 0 to 10)*										
1	randomised trials	serious ^a^	not serious	not serious	serious ^b^	none	10	10	SMD **1.24 SD** **lower** (2.21 lower to 0.26 lower)	⨁⨁◯◯ Low
*Temporalis Right (follow-up: mean 3 weeks; assessed with: EMG; Scale from: 0 to 10)*										
1	randomised trials	serious ^a^	not serious	not serious	serious ^b^	none	10	10	SMD **1.06 SD** **lower** (2.01 lower to 0.11 lower)	⨁⨁◯◯ Low
*Temporalis Left (follow-up: mean 3 weeks; assessed with: EMG; Scale from 0 to 10)*										
1	randomised trials	serious ^a^	not serious	not serious	serious ^b^	none	10	10	SMD **1.16 SD** **lower** (2.12 lower to 0.19 lower)	⨁⨁◯◯ Low

**CI:** confidence interval; **SMD:** standardized mean difference; ^a^ the study present “Some concerns” in RoB2; ^b^ the number of the total participants were under 300; ^c^ the CI crossed the line. ⨁◯◯◯ = very low evidence level; ⨁⨁◯◯ = low evidence level; ⨁⨁⨁◯ = moderate evidence level; ⨁⨁⨁⨁ = high evidence level.

## Data Availability

Not applicable.

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
