# Peer review of "Effectiveness of Biofeedback in Individuals with Awake Bruxism Compared to Other Types of Treatment: A Systematic Review"

_ijerph, 2023, doi:10.3390/ijerph20021558_

Round 1

Reviewer 1 Report

This is a nice review on biofeedback in daytime bruxism patients.

I have only some marginal comments. The manuscript is well written and clear.

An overall statement is that the citations should be checked carefully for duplicates. This is rather difficult, since some papers appear with different names; see citation 23 and 38 as example.

To my opinion, the introduction can be shortened by removing a few duplications. But only marginally, it reads well.

I wonder why studies with elderly patients (>60y) and with gastric reflux are actively excluded. With daytime bruxism, reflux is hardly a problem and 60 yrs is these days relatively young.

Line 216 and figure 1: Please check the numbers. I calculated 80 reports removed based on the numbers provided. But also the numbers in the square “Records identified from” do not sum op to 4059, nor do the removed records sum up correctly. I do not know where the 6 records in “Reports not retrieved” come from.  And 4,059 minus 2,339 gives 1,720 and not 1,664 as given in “records screened”. So I miss several records. Please take care that the numbers are correct.

Line 218: The percentages are not needed here. Moreover, 23.17% of 82 is far too specific, so take care that the numbers of significant digits are correct (23% in this example).

I completely miss a statement about the fact that 3 from the 4 included studies are from one and the same department (Meikai University School of Dentistry, Sakado, Japan). This is a high risk of bias, since systematic mistakes can be made easily is such cases. This should be included in the report.

Line 357: The reverse can also be possible. That the level of stress can affect the bruxism. This should be noted.

Table 2 and 3 are absolutely unreadable in the current layout and need to be adjusted.

I think that the appendix should not be placed between the manuscript and the reference list.

Author Response

Review Letter

“Effectiveness of biofeedback in individuals with awake bruxism compared to other types of treatment: A systematic review”

Reviewer 1

Response

1.  An overall statement is that the citations should be checked carefully for duplicates. This is rather difficult, since some papers appear with different names; see citation 23 and 38 as example.

Thank you for your comment. We agree that some references were duplicated. We have double-checked the references and updated our software manager to correct this.

2. To my opinion, the introduction can be shortened by removing a few duplications. But only marginally, it reads well.

We reviewed the introduction to reduce the redundancy of information. We tried to reduce the text as much as possible without losing the point.

All the changes that we have made in the introduction are highlighted in yellow in the manuscript.

I wonder why studies with elderly patients (>60y) and with gastric reflux are actively excluded. With daytime bruxism, reflux is hardly a problem and 60 yrs is these days relatively young.

The inclusion and exclusion criteria of the paper were chosen based on the prevalence of the condition in the population. We just included adults because of the higher prevalence of bruxism in that population. (1-3) But it is important to highlight that we did not find any study with a patient group with more than 60 years old, thus this was not a relevant reason to exclude a paper.

About the reflux, we corrected our sentence since we excluded studies that just included gastric reflux without a bruxism component. If the patient presented the combination, they would be included. But we highlight we did not find any study with this clinical conditions combination.(4)

We added this sentence in the text:

”gastroesophageal reflux exclusively“ (Page 3, Line 143)

Line 216 and figure 1: Please check the numbers. I calculated 80 reports removed based on the numbers provided. But also the numbers in the square “Records identified from” do not sum up to 4059, nor do the removed records sum up correctly. I do not know where the 6 records in “Reports not retrieved” come from.  And 4,059 minus 2,339 gives 1,720 and not 1,664 as given in “records screened”. So I miss several records. Please take care that the numbers are correct.

Thank you for the observation. Since some incompatibilities were found, we did a double-checked review of these numbers and updated figure 1.

Here we describe the number of papers just to be clearer:

In total there were 4,059 results, provided by the following databases: OVID/Medline (n =1,051); Embase (n=416); Pubmed (n=792); Web of Science (n=1,579); Cochrane Library (n=162); Cinahl (n=59). After the removal of duplicates by Endnote and Covidence (total = 2,395), there were 1,664 records screened. After reading title and abstract, 86 studies were eligible for full reading. Then, based on the inclusion and exclusion criteria only 4 studies remain this review.

Line 218: The percentages are not needed here. Moreover, 23.17% of 82 is far too specific, so take care that the numbers of significant digits are correct (23% in this example).

We agree that the percentages are not mandatory, but we think that it is better for reader to have the proportional results instead of to have just the general numbers, so we decided to provide them. But in order to make this more readably, we changed all the specific percentages to a more readable percentage, without decimals values, as recommended by the reviewer.

I completely miss a statement about the fact that 3 from the 4 included studies are from one and the same department (Meikai University School of Dentistry, Sakado, Japan). This is a high risk of bias, since systematic mistakes can be made easily is such cases. This should be included in the report.

We agree with this statement, but because all the studies published accomplished with the inclusion and exclusion criteria from our review, we included them. But we highlighted this potential risk of bias in the discussion section (4.6. Strength and limitations).

Line 357: The reverse can also be possible. That the level of stress can affect the bruxism. This should be noted.

In order to accomplish this statement, we rewritten this sentence in the text:

Thus, when biofeedback therapy has a positive effect on reducing masticatory activity, it is supposed that in consequence biofeedback technique promotes the relaxation of tense masticatory muscles through awareness of the habit of clenching teeth and the reduction of stress (emotional) level, which can directly affect the symptoms of bruxism, which are closely related to stress levels (Lines 355 – 359)

Table 2 and 3 are absolutely unreadable in the current layout and need to be adjusted.

We apologize for this. We updated the tables to have a better layout.

I think that the appendix should not be placed between the manuscript and the reference list.

Thank you for the observation. We deleted the appendix in the text and add as an independent document.

References

  1. Hilgenberg-Sydney PB, Lorenzon AL, Pimentel G, Petterle RR, Bonotto D. Probable awake bruxism-prevalence and associated factors: a cross-sectional study. Dental Press Journal of Orthodontics. 2022;27.

  1. Manfredini D, Winocur E, Guarda-Nardini L, Paesani D, Lobbezoo F. Epidemiology of bruxism in adults: a systematic review of the literature. Journal of orofacial pain. 2013;27(2):99-110. doi:10.11607/jop.921.

  1. Peixoto KO, Resende CMBMd, Almeida EOd, Almeida-Leite CM, Conti PCR, Barbosa GAS, Barbosa JS. Association of sleep quality and psychological aspects with reports of bruxism and TMD in Brazilian dentists during the COVID-19 pandemic. Journal of Applied Oral Science. 2021;29.

Reviewer 2 Report

Page 2, lines 103–107: The authors mentioned in the introduction that previous systematic reviews had limited scope because those do not include a wide range of databases. Can the authors comment on the risk of bias in the studies they included for their analyses (maybe in the discussion section)? Even this systematic review does not include grey literature to an expanded extent. For instance, they did not search the ProQuest thesis repository or other similar databases. Also, it is not clear in the text whether conference proceedings/abstracts were also searched.
Page 3, lines 118-122: was the age limit set because the trials included had it as their inclusion criteria, or did the authors set it for the ease of their data collection? Again, the effective definition of older adults may vary depending on many factors; how did the authors homogenize that?
Page 5, line 215–2016: I see the number of studies are in fractions—perhaps because commas are replaced with full stops. Please double-check this.
Page 11, lines 274–279: Apparently, the overall rob of the studies was pretty high. This makes the whole study's outcome controvertible. Perhaps including more studies would be helpful. I suggest they search more databases and try to include more studies in this meta-analysis.
Page 12, section 3.4: Could the authors identify a sex-biased effect?
Page 13, lines 321–323: Please tabulate these data in the supplementary data as well, as they may be helpful for some readers.
Given that (i) only four studies were included, (ii) the level of bias in those included studies was high, and (iii) there are previous systematic reviews and meta-analyses on this topic, this reviewer is not convinced that the study has significant novelty. However, the author may wish to provide more rationale on why their study is essential.

Author Response

Review Letter

“Effectiveness of biofeedback in individuals with awake bruxism compared to other types of treatment: A systematic review”

Reviewer 2

Response

Page 2, lines 103–107: The authors mentioned in the introduction that previous systematic reviews had limited scope because those do not include a wide range of databases. Can the authors comment on the risk of bias in the studies they included for their analyses (maybe in the discussion section)? Even this systematic review does not include grey literature to an expanded extent. For instance, they did not search the ProQuest thesis repository or other similar databases. Also, it is not clear in the text whether conference proceedings/abstracts were also searched.

We discussed these aspects of the limitation on the previous systematic reviews in the discussion section.

Also we would like to reinforce that we did use enough number of database to answer our questions, all of them are highlight in the methods sections:

“The searches were conducted by a team (MAV, LB, GH, AISOS) on October 1, 2021 and updated on July 12, 2022, in the following five databases: EBSCO/ CINAHL, Embase, PubMed/ Medline, Cochrane library (Wiley Interface), Web of Science (Indexes=SCI-EXPANDED, SSCI, A& HCI, ESCI). (Page 3, Lines 103 – 107)

And we also looked into the conference proceedings/abstracts, however just the full texts published were included in this review:

“Two independent reviewers screened the titles and abstracts in the first step and then full texts considering previously established inclusion and exclusion criteria. (Page 4, Lines 155 – 157)

Page 3, lines 118-122: was the age limit set because the trials included had it as their inclusion criteria, or did the authors set it for the ease of their data collection? Again, the effective definition of older adults may vary depending on many factors; how did the authors homogenize that?

We choose this age limit because of higher percentil of awake bruxism in this population group. (1–3) We based the elderly definition on the definition provided by the World Health Organization (WHO), where elderly is every individual aged 60 years or more. (5,6)

Page 5, line 215–2016: I see the number of studies are in fractions—perhaps because commas are replaced with full stops. Please double-check this.

We doubled-checked all the numbers that we provided in the text to avoid misspelling.

Page 11, lines 274–279: Apparently, the overall rob of the studies was pretty high. This makes the whole study's outcome controvertible. Perhaps including more studies would be helpful. I suggest they search more databases and try to include more studies in this meta-analysis.

We understand and agree there are few studies in our systematic review. Unfortunately, it is not a limitation regard to our systematic review, but related to the real published literature where there are just few studies about biofeedback and awake bruxism. We would like to reinforce that we already included the most important databases in our searches, suggested by the Cochrane Collaboration as the essential databases to be searched (i.e. EBSCO/ CINAHL, Embase, PubMed/ Medline, Cochrane library (Wiley Interface), Web of Science (Indexes=SCI-EXPANDED, SSCI, A& HCI, ESCI ).

Page 12, section 3.4: Could the authors identify a sex-biased effect?

Unfortunately, we do not have enough studies with different sex/gender groups, that allowed us to analyze sex-bias effect in this systematic review. Two studies included both sexes in the groups (7,8); and the other two include only man in the studies. (9,10)

Page 13, lines 321–323: Please tabulate these data in the supplementary data as well, as they may be helpful for some readers

Thank you for the information. We add the appendix as a supplementary document.

Line 357: The reverse can also be possible. That the level of stress can affect the bruxism. This should be noted.

In order to accomplish this statement, we rewritten this sentence in the text:

Thus, when biofeedback therapy has a positive effect on reducing masticatory activity, it is supposed that in consequence biofeedback technique promotes the relaxation of tense masticatory muscles through awareness of the habit of clenching teeth and the reduction of stress (emotional) level, which can directly affect the symptoms of bruxism, which are closely related to stress levels (Lines 355 – 359)

Given that (i) only four studies were included, (ii) the level of bias in those included studies was high, and (iii) there are previous systematic reviews and meta-analyses on this topic, this reviewer is not convinced that the study has significant novelty. However, the author may wish to provide more rationale on why their study is essential.

Although there are systematic reviews on this topic, no review has ever attempted to explain the benefits of biofeedback in patients with daytime bruxism, and therefore this review is extremely necessary, to clarify what has been done in this topic until nowadays.

The high risk of bias in the studies included in this review reinforces the important recommendation that new clinical trials with good methodological quality should be conducted in order to better identify these results.

We reinforced this in the section about “Considerations for research and clinical practice” at the discussion part.

References

  1. Peixoto KO, Resende CMBM de, Almeida EO de, Almeida-Leite CM, Conti PCR, Barbosa GAS, et al. Association of sleep quality and psychological aspects with reports of bruxismand TMD in Brazilian dentists during the COVID-19 pandemic. J Appl Oral Sci. 2021;29:e20201089.
  2. Hilgenberg-Sydney PB, Lorenzon AL, Pimentel G, Petterle RR, Bonotto D. Probable awake bruxism - prevalence and associated factors: a cross-sectionalstudy. Dental Press J Orthod. 2022;27(4):e2220298.
  3. Manfredini D, Winocur E, Guarda-Nardini L, Paesani D, Lobbezoo F. Epidemiology of bruxism in adults: a systematic review of the literature. J Orofac Pain. 2013;27(2):99–110.
  4. Lobbezoo F, Ahlberg J, Glaros AG, Kato T, Koyano K, Lavigne GJ, et al. Bruxism defined and graded: an international consensus. J Oral Rehabil. 2013 Jan;40(1):2–4.
  5. Beard JR, Officer A, de Carvalho IA, Sadana R, Pot AM, Michel J-P, et al. The World report on ageing and health: a policy framework for healthy ageing. Lancet (London, England). 2016 May;387(10033):2145–54.
  6. Kalache A, Gatti A. Active ageing: a policy framework. Adv Gerontol = Uspekhi Gerontol. 2003;11:7–18.
  7. Wieselmann-Penkner K, Janda M, Lorenzoni M, Polansky R. A comparison of the muscular relaxation effect of TENS and EMG-biofeedback inpatients with bruxism. J Oral Rehabil. 2001 Sep;28(9):849–53.
  8. Watanabe A, Kanemura K, Tanabe N, Fujisawa M. Effect of electromyogram biofeedback on daytime clenching behavior in subjects with masticatory muscle pain. J Prosthodont Res [Internet]. 2011;55(2):75–81. Available from: http://dx.doi.org/10.1016/j.jpor.2010.09.003
  9. Saito‐Murakami K, Sato M, Otsuka H, Miura H, Terada N, Fujisawa M. Daytime masticatory muscle electromyography biofeedback regulates the phasic component of sleep bruxism. J Oral Rehabil [Internet]. 2020 Jul;47(7):827–33. Available from: http://search.ebscohost.com/login.aspx?direct=true&db=c8h&AN=143823342&amp
  10. Sato M, Iizuka T, Watanabe A, Iwase N, Otsuka H, Terada N, et al. Electromyogram biofeedback training for daytime clenching and its effect on sleep bruxism. J Oral Rehabil. 2015;42(2):83–9.

Round 2

Reviewer 2 Report

This reviewer is still not convinced that the limitations of the previous studies are adequately described to provide a strong rationale for the current study. This is a stand-alone new manuscript; whatever was discussed in earlier publications is not part of this study.

Systematic reviews rely on the availability of meta-data; the lack of the availability of quality data does affect the conclusions from a systematic review. The Inadequateness of a systematic review may mislead the reader, which is why it is essential to be careful when drawing conclusions from metadata with inadequacies.

The authors need to address these issues explicitly in the manuscript.

Author Response

Review Letter

“Effectiveness of biofeedback in individuals with awake bruxism compared to other types of treatment: A systematic review”

Reviewer 2

Response

1.  This reviewer is still not convinced that the limitations of the previous studies are adequately described to provide a strong rationale for the current study. This is a stand-alone new manuscript; whatever was discussed in earlier publications is not part of this study.

The main objective of this review was to compile and synthesize the information about the effectiveness of biofeedback for managing “awake bruxism” in adults, compared to other types of treatments.

The population of interest for our review was patients with awake bruxism since previous reviews did NOT address and focus on this specific population. As highlighted in our introduction, previous systematic reviews have been published to elucidate the effects of physiotherapy treatments on sleep bruxism, however, they have not included adults with awake bruxism. Therefore, the use of biofeedback on this population in specific has not been addressed in the existing literature, and this comprises the scientific gap which supports the need for a review that summarizes this literature. Furthermore, in those previous reviews that have included bruxism in general (without specifying the diagnosis), authors have combined both types of bruxism. Thus it was not possible to determine the effectiveness of this therapy separately for both diagnoses. Also, some methodological issues were found in all previous published systematic reviews, such as limitations due to language bias, search strategies issues, few databases included, and data analyses, which directly affect the quality of the existing evidence in these previous systematic reviews, and consequently, create a bias in the results. We have highlighted these weaknesses in our introduction to highlight the need for this review.

Our review has the objective to improve these methodological aspects focusing on awake bruxism. We have tried to make this information clear throughout the introduction of the manuscript and also in parts of the discussion.

2. Systematic reviews rely on the availability of meta-data; the lack of the availability of quality data does affect the conclusions from a systematic review. The Inadequateness of a systematic review may mislead the reader, which is why it is essential to be careful when drawing conclusions from metadata with inadequacies.

We agree with the reviewer that the conclusions of a review are affected by the quality of the data included in the review. However, the quality of a review itself depends on its methodology and how the process was conducted. We have followed a strict methodology to search, select, and analyze included studies. Although we included a few studies in our review, this was not due to a poorly conducted review, rather, our review points out that this area needs to be further developed. We want to mention that our search strategy has been developed with the help of an expert librarian and that the most important databases have been included in our search as recommended by the leading groups conducting systematic reviews (Cochrane Collaboration).

We have changed our conclusion to be more cautious when presenting our results in agreement with the reviewer’s comments. (Page: 19; Line 455 - 466)

3. The authors need to address these issues explicitly in the manuscript.

As we reported above, we have addressed the issues raised by the reviewers in the best way we could. All the changes made in the text are highlighted in light blue in the manuscript.